# GIS Models for Vulnerability of Coastal Erosion Assessment in a Tropical Protected Area

**Luís Russo Vieira [1,2], José Guilherme Vieira [3]**, **Isabel Marques da Silva [4], Edison Barbieri [5]** and **Fernando Morgado [3],\***

1   Interdisciplinary Centre of Marine and Environmental Research (CIIMAR), University of Porto, Terminal de Cruzeiros do Porto de Leixões, Av. General Norton de Matos s/n, 4450-208 Matosinhos, Portugal; bioluis@ciimar.up.pt
2   ICBAS Institute of Biomedical Sciences of Abel Salazar, University of Porto, Rua de Jorge Viterbo Ferreira n.° 228, 4050-313 Porto, Portugal
3   Department of Biology & CESAM, University of Aveiro, Campus Universitário de Santiago, 3810-193 Aveiro, Portugal; jose.guilherme.vieira@ua.pt
4   Faculty of Natural Sciences, Lúrio University, 958 Pemba, Mozambique; isilva@fcn-unilurio.com
5   Fisheries Institute APTA-SAASP, Government of Sao Paulo State, Postal Code 157, Cananeia 11990-000, Brazil; edison.barbieri@sp.gov.br
*   Correspondence: fmorgado@ua.pt

**Abstract:** Coastal erosion is considered a major worldwide challenge. The vulnerability assessment of coastal areas, in relation to climate change, is a key topic of worldwide increasing interest. The integration of methodologies supported by Remote Sensing, Geographical Information Systems (GIS) and in situ monitoring has allowed a viable identification of vulnerable areas to erosion. In the present study, a model was proposed to the assessment of the estuarine system of Cananéia-Iguape (Brazil), by applying the evaluation and prediction of vulnerability models for the conservation and preservation of mangroves. Approximately 1221 Km$^2$ were classified, with 16% of the total presenting high and very high vulnerability to erosion. Other relevant aspects, were the identification and georeferencing sites that showed strong evidence of erosion and, thus, having a huge influence on the final vulnerability scores. The obtained results led to the development of a multidisciplinary approach through the application of a prediction and description model that resulted from the adaptation of the study system from a set of implemented models for coastal regions, in order to contribute to the erosion vulnerability assessment in the mangroves ecosystems (and associated localities, municipalities and communities).

**Keywords:** remote sensing; coastal erosion; mangroves; Geographical Information Systems; coastal vulnerability index

## 1. Introduction

Coastal erosion is becoming an increasingly severe problem for worldwide coastal ecosystems, derived from coupled impacts of climate change, through sea-level rise, and intensified anthropogenic activities. Coastal erosion takes place mainly during strong winds, high waves and high tides and storm surge conditions, and results in coastline retreat and loss of land. Due to the potential future changes that could occur with regards to human interference, and increased storm occurrence/severity, sea-level rise, and wave climate as a result of climate change [1,2] any prediction about future coastal erosion rates as they vary across locations is problematic. The losses resulting from this phenomenon are enormous and affect not only the environment but also the set of human activities and structures [3,4], considering that more than 20% of the world's population live within 25 km from the coastline [5]. These impacting processes had led to cumulative negative effects with several consequences in adjacent coastal areas, including (i) coastline retreat; (ii) disappearance of beach areas; (iii) loss and imbalance of natural habitats; (iv) increase in

the frequency and magnitude of floods; (v) loss of public and private property and assets; (vi) loss of landscape value; (vii) severe damage to socio-economic activities; (viii) an on the recovery of coastal areas [6–8]. Therefore, managing coastal erosion under climate change is a major need for reliable projections of shoreline change across time scales up to multidecadal and centennial [5].

The role of vegetation in shoreline stabilization is being increasingly recognised [9]. In particular environmental scientific community often assign a high value to tropical mangroves in coastline protection [9–11]. Mangroves are outstanding land-sea interphase forest ecosystems, comprised by tropical and subtropical plants. They are characterized to have their roots partly or wholly submerged in the land-sea interphase waters [2,11]. They are amongst the most productive and carbon-rich ecosystems on Earth, being widely distributed and can be found in both tropical and subtropical areas, where topographic and physical conditions of the substrate are favourable to their establishment [6,12,13]. These type of interphase ecosystems are, therefore, subject to several instabilities that vary in their intrinsic nature, including chemical, biological, geological and physical, in space and time; for example, they are continuously subjected to tidal changes in temperature, water and salt exposure, and varying degrees of anoxia [14]. Living at the interface between land and sea, these forests have been recently considered sentinels for climate change [15,16]. Additionally, these interface ecosystems provide an important and wide range of ecosystem services, including coastal protection, carbon sequestration and opportunities for biodiversity [17–19]. Even considering the key role of Mangrove forests as ecosystems services providers and bioindicators for multiple anthropogenic effects, these are feasibly the most undervalued and trivialized ecosystems in the world [13,16]. Anthropogenic activities and climate change impacts on these interphase forests have received increasing concern and attention, especially because the alarming mangrove deforestation, occurring at an average rate of 1–2% per year, being even more drastic in developing countries, which implies that, without robust sustainable policies, most of these ecosystems will disappear within this century [7,14]. More than 35% of the total existing mangroves forests have been lost over the last 30 years [7]. In addition, more than 40% of mangrove plant species are listed as Threatened on the International Union for Conservation of Nature (IUCN) Red List [13]. The decline of these forests is mainly due to the destruction of mangroves for the construction of urbanizations, agricultural fields or infrastructure, which in turn are affecting other adjacent areas due to their embrittlement caused by human action [20–22]. As a major consequence erosion is considered to be one of the major problems of today's coastal zones and is of extreme importance for decision-making, assessing and predicting risks in order to avoid losses [22].

Considering these concerning issues and scenarios, there is an urgent need to monitor the areas where these impacting processes take place. In order to effectively study these ecosystems and to integrate in situ monitoring (including mangrove forests changes overtime), accurate, timely, and cost-effective mapping techniques are urgently required [4,7]. Considering the unstable environment of mangrove forests, the Remote Sensing (RS), associated with Geographic Information Systems (GIS), has been widely proven to be an essential and sustainable tool for monitoring and mapping these ecosystems [7,23]. Using these types of visual representations technologies, using data from different years, it is possible to evaluate the changes occurring in any region [10,24–26]. The use of these representations with the appropriate software is complemented by a method that leads to a better understanding of the spatial distribution of destructive processes [3,26,27]. Given this framework, there is a need to predict the behaviour of erosive processes in order to prevent such damages [27]. The study application in this area of research has been increasing during the last years, being possible to identify a significant variety of methodologies with different types of application that not only identify the most vulnerable areas but also calculate the erosion rates [7,26]. The tools provided by GIS allow several combinations that can be used to solve a myriad of problems involving spatial data [28]. The main advantage of using GIS, combined with other methodologies, is their ability to store, modify

and retrieve large sets of data of heterogeneous origin and to represent them in a visual format [24,29]. The use of these tools depends on the quality, availability and accuracy of the data, and it is necessary to take these aspects into account in order to maintain a solid scientific base so as not to jeopardize any work coming from such sources of GIS to evaluate, map and quantify the set of variables that contribute to the erosion of risk areas is essential to understand the dynamics of mangroves [24,30].

Several attempts to use the Universal Soil Loss Equation in different regions as a technique for predicting erosion rates and evaluating different soil conservation practices show the need for special care in the input values of some variables [31]. In addition, the equation is of limited value, since it does not provide information on sediment fate, which occurs during erosive processes [31]. Given all this information feedback, several models have been conducted to evaluate erosion prediction under different soil management conditions, as well as to adapt to user needs. The European Soil Erosion Model (EUROSEM) is a dynamic distribution model capable of simulating the transport of sediment, erosion and deposition on the land surface, as discussed by [31], describing the characteristics that make it different from the other models, emphasizing that for some countries the most important consequence of erosion is sedimentation downstream rather than loss of local productivity, so it is necessary to apply this model or similar to obtain such results. [32] used the European Soil Erosion Model to evaluate data from individual events in localities in China and report that, according to their results, EUROSEM surface runoff reasonably well, it is not able to simulate the concentration of sediments and the rate of soil loss with precision in a single event; however, it is able to differentiate the impacts of land use and soil protection measures on runoff, as well as total soil loss. Many authors have opted for cognitive approaches to qualitative models, in order to better understand the spatial distribution of erosion [33]. This is feasible due to the possibility of selecting a set of parameters that best approximate the reality before the erosive processes.

The estuary system Cananéia-Iguape is one of the most important wetland areas of the Brazilian coast in terms of biodiversity and natural productivity. This is recognized nationally and internationally as the third most productive ecosystem in the South Atlantic [34]. For this reason, it is considered the Atlantic Forest Reserve of the Biosphere since 1993 and the World Natural Heritage of scientific knowledge and preservation of human values and traditional knowledge due to the good preservation of its environmental characteristics [34]. Despite the mangroves of the South Coast of São Paulo being considered the most preserved of the state, these present clearings in mangrove areas, and a lot of invasive aquatic weeds [35]. This region, like many others in the south coast of São Paulo, has several protected areas due to their environmental relevance and importance, the habitat for marine and estuarine species, and in this region, can be found dozens of islands, mangroves in good condition, affluence of unpolluted small rivers and relatively small human settlements, and thus ensure the natural attributes to this region [36]. Although these coastal systems are considered as very important areas of the Brazilian coast, human activities have had significant impacts on the estuary system of Cananéia-Iguape, emphasizing the effects of the opening of the artificial channel also changed the sedimentation patterns [35]. The set of all these factors contributes to an increased vulnerability of erosion in this area and is necessary to identify the criteria that better reflect this process, and the need to predict the behaviour of erosion to prevent its losses [27].

In the present study, it was developed a model for an adequate management of the estuarine system of Cananéia-Iguape, by applying evaluation and prediction of vulnerability models for the conservation and preservation of mangroves. The variety of coastal environments hinders the creation of a single model to apply in various mangrove areas because of the disproportionate concentration in coastal areas cause impacts at different scales compromising the generalization of methodologies [37,38]. The approach of a global methodology for vulnerability identification in coastal regions tried to consider the parameters that best characterize the erosion of the study area, such as physical parameters and also socio-economic, demographic and economic in order to achieve a better char-

acterization of global vulnerability [37]. The main objective of the present work was the development of a multidisciplinary information approach by adapting the study system to a validated set of models of coastal systems in other regions. The application of the GIS model was adapted and applied to the complexity of the mangrove estuarine system of Cananéia-Iguape, (Sao Paulo, Brazil) to determine the vulnerability of coastal erosion. Specifically, the application of the GIS model included the (i) study area characterization; (ii) identification of vulnerability parameters; (iii) creation of vulnerability maps; (iv) combination of the several maps; (v) creation of the global vulnerability map of the study area; (vi) review the results obtained.

## 2. Materials and Methods

### 2.1. Study Area

The present research was developed in the Cananéia-Iguape estuarine system, located in the southern littoral mesoregion from the State of São Paulo coast, between at latitude between 24°50′ and 25°40′ South and longitude between 47°20′ and 48°20′ West (Figure 1). It is composed of complex environments associated with barrier islands, mangroves, lagoon channels, muddy plains and marshes [39]. In the region of Cananéia, is located the main system channel (Mar de Cananéia), with a width of not more than 1 km and of approximately 75 km in length, parallel to Ilha Comprida and with greater depth next to the bar of Cananéia (6 a 7 m) (Figure 1) [40]. This system has an area of 1434 km$^2$, being limited to the north by the municipality of Iguape, Ilha Comprida, on the west by the Serra do Mar and on the south by the islands of Cananéia and Cardoso, which are formed mainly by large unconsolidated sediments and metamorphic rocks. It is also linked to the ocean by two connections: to the north, through a single channel (Barra de Icapara) and to the south through two channels (Barra de Cananéia and Ararapira). These islands are separated by rivers and lagoons communicating with the Atlantic Ocean. The land surface is covered mostly by Rain and Halophyte Forest (approximately 83%), highlighting the mangrove of Cananeia, and its affluent. The climate is characterized according to the Brazilian Institute of Geography and Statistics as subtropical with warm temperatures and constant periodic rain over the year. The study area has a great number of important protected areas; although they have these designations with protective status, it is very common to see the exploitation of resources provided by the mangrove of Cananeia. According to the Koppen climatic classification, the region climate is considered subtropical humid, mesothermic, with hot summer's characteristic, with a tendency to high rainfall concentrations at this time, with no defined dry season and with periods of frost infrequent [18]. The average annual precipitation is 2300 mm, which is well distributed throughout the year, and in the summer quarter (December, January and February) it reaches a monthly average of 266.9 mm and in the winter quarter (June, July and August) round around a monthly average of 95.3 mm [18].

The circulation within the system is mainly driven by the action of the tidal waves, which enter through the Cananéia, Ararapira, Icapara, and freshwater contributions of several rivers of the region, still suffering from wind influences [41]. According to [42], the average height of the tide, recorded at the Cananéia base, is 81 cm, and in tandem and quadrature tides reaches values of 120 and 26 cm respectively. In the city of Iguape, a canal (Valo Grande) was constructed with the aim of facilitating navigation at the end of the Ribeira River, presenting a width of 4.40 m after its construction [39]. After this construction, the estuarine-lagoon system of Cananéia Iguape has undergone changes in its dynamics [39]. At present, as a consequence of coastal erosion, the margins of this area have been degraded, being the channel with more than 300 m of width, causing much of the flow of the river Ribeira to flow through the Valo Grande leading to a decrease in the salinity in the system estuarine-lagoon [43]. Years after the construction of the Valo Grande, the Government of the State of São Paulo decided to close the Valo for the construction of a dam, again inciting changes in the ecosystem [44]. With the rupture of the dam, it triggered

changes at the hydrodynamic level of the system with the intensification of the currents and an increase in the salinity [44].

**Figure 1.** Location of the study area, the estuarine system of Cananeia-Iguape at the extreme south of São Paulo coast, Brazil. The areas vulnerable to erosion are identified.

### 2.2. The Mangrove Forest of the Cananéia-Iguape Estuarine-Lagoon System

This type of estuarine system is characterized by the black mangrove (*Avicennia Schaueriana*), the red mangrove (*Rhizophora mangle*) and the white mangrove (*Laguncularia racemosa*) [45]. In this region Spartina's growth in swampy marshes during the summer is very common; the leaves and roots of this genus of plants provide shelter for the animal community, which is dominated by isopods, amphipods, polychaetes, gastropods, bivalves and decapods [45]. Most of the animal populations follow the cycle of these grasses, when it reaches maximum vegetation cover formation in autumn, or maximum dead blanket formation in winter, supporting a larger and more diverse community [45]. According to [34], the estuarine-lagoon system of Cananéia-Iguape is one of the most important wetlands of the Brazilian coast in terms of biodiversity and natural productivity. It is recognized nationally and internationally as the third most productive ecosystem of the South Atlantic, being considered as a Biosphere Reserve of the Atlantic Forest in 1993, as well as a Natural World Heritage Site, scientific knowledge and the preservation of human values and traditional knowledge with sustainable development models, due to the good preservation of their environmental characteristics [34]. This region, like many of the south coast of São Paulo, has several protected areas due to its environmental relevance and importance as a habitat for marine and estuarine species. It is possible to find in this region, dozens of islands, mangroves in good state of preservation, affluence of small unpolluted rivers and a relatively small human occupation, thus guaranteeing the natural attributes to this region [36]. In the study area, there are several state and federal units that vary in terms of the degree of use restriction, as well as in the aptitude for its management, highlighting the Environmental Preservation Area of Cananéia, Iguape and Peruíbe, State Environmental Preservation Area of Ilha Comprida, Tupiniquins Ecological Station and Areas of Relevant Ecological Interest of Queimada Grande and Queimada Pequena, Ecological Station of Juréia-Itatins, Ecological Station of Chauás, State Park of

Cardoso Island, State Park of Jacupiranga and Extractivist Reserve of Mandira SEMASP 1990 [36].

### 2.3. Field Methodology

The field methodology included; (i) data collection; (ii) vulnerability parameters selection; (iii) vulnerability parameters maps; (iv) creation of global vulnerability index and (v) global vulnerability map of the estuarine system of Cananéia-Iguape. The GIS consisted in the specialization of the global vulnerability of coastal erosion in the region, through the application of a qualitative method, based on a cognitive approach, having been set a number of factors in terms of sensitive categories defined from validated models such as ICZM (Integrated Coastal Zone Management), CVI (Coastal Vulnerability Index), CALOD Index (Clay layer thickness (C), Aquifer media character (A), Lateritic layer thickness (L), Overlaying layer character (O) and the Depth to groundwater level (D)) and LTC (Long Term Configuration) [46–51].

Prior to the model development for the spatial classification of the Cananeia coastal system vulnerability, it was implemented an identification of the sites which presented strong evidence of erosion, such as areas with infrastructures the presented evidence of human activity which directly influence erosion or by the natural dynamic of estuary system, through the development of fieldwork to record the coordinated and subsequent georeferencing areas mostly affected by this process. This record was done between December 2013 and February 2014 and resulted in the elaboration of a map of areas vulnerable to erosion (Figure 1). The remaining data were collected at the Water Resources Fund of Sao Paulo [52].

### 2.4. Vulnerability Parameters and Maps

The analysis of the vulnerability of coastal areas is usually based on the parameters of Geology, Geomorphology and topographic elevation [49,51,53]. Due to the complexity of the coastal system of Cananéia, it was necessary to perform a detailed analysis to select the vulnerability parameters that could better feature the study area. Seven vulnerability parameters were selected, containing quantitative and qualitative information, defined and classified individually. Due to the different dimensions of each parameter, it was necessary to classify vulnerability which ranged from 1 (very low) to 5 (very high), defined according to [48,49,51,53] (Table 1). Regarding the relevance of the parameters, it stands out the Anthropogenic Activities (AA) corresponding to the activities and anthropogenic processes along with the coastal areas that influence their natural dynamics. In spite of mangroves being considered a good way to protect against erosion, in the study area, it was very common to find practices such as deforestation, exploitation of natural resources or construction of infrastructure that caused interference in the natural processes of the region [12,54]. The Land Cover (LC) was also considered as a very important parameter, because it tends to vary over time, increasing more often the number and size of vulnerable areas. For the remaining parameters, they corresponded to the natural attributes of the study area and to human intervention: Elevation (E) distinguishes the coastal plains from large slope areas; Geomorphology (GM) distinguishes the different areas such as mountain and dunes; Geology (GO) differentiates the rock types such as unconsolidated sediments and magmatic rocks, being classified according to their resistance against erosive agents; the Distance to the Coastline (DC) reflected the vulnerability depending on approximation to the coastline; the Maximum Tidal Range (MA), a phenomenon characterized by periodic increases and lowering's of the sea being presented an average level of 2.20 m [40]. The development of vulnerability maps made possible the creation of a tool that made it easier to preview and categorize the study area. These maps helped to delimitate the areas according to their vulnerability. For each parameter, it was made a vulnerability map and each one of them was presented in five vulnerability levels.

**Table 1.** Defined parameters for the study area, being each attribute of each parameter associated with a vulnerability category [48,51,55,56].

| Parameter/Vulnerability | 1-Very Low | 2-Low | 3-Medium | 4-High | 5-Very High |
|---|---|---|---|---|---|
| Elevation (m) | >30 | 20 to 30 | 10 to 20 | 5 to 10 | <5 |
| Geomorphology | Mountains | Rocky cliffs | Saltwater marshes Mangroves Coral reefs Sheltered beaches | Floodplains Exposed beaches Estuaries | Dunes |
| Geology | Magmatic rocks | Metamorphic rocks | Sedimentary rocks | Large unconsolidated sediments | Small unconsolidated sediments |
| Land cover | Forest | Undergrowth, crops | Soil without covering | Rural urbanization | Urbanization |
| Anthropogenic activities | Interventions with maintenance structures in the coastline | Interventions without structures, but without evidence sedimentary reduction | Interventions without structures, but with evidence of sedimentary reduction | Without interventions and with no evidence of sedimentary reduction | Without interventions, but with evidence of sedimentary reduction |
| Distance to the coastline (m) | >1000 | 200 to 1000 | 50 to 200 | 20 to 50 | <20 |
| Maximum tidal range (m) | <1 | 1 to 2 | 2 to 4 | 4 to 6 | >6 |

*2.5. Global Vulnerability Index*

After processing all parameters in standard vulnerability maps, it was carried out a combination of them, to determine the overall vulnerability of the study area. According to [48], two different approaches can be used to access the overall vulnerability index, (i) by multiplying the parameters where $x$ represents the parameter and n the total number of parameters,

$$CVI = \left[\frac{1}{n}(x1 \times x2 \times \dots xn)\right]^{\frac{1}{2}} \qquad (1)$$

$x$—parameter; $n$—total number of parameters or (ii) using the sum of the variables, where X represents the individual parameter and N is the assigned weight. The multiplication has the advantage of increasing the range values, on the other hand, the results may vary with small changes in the individual classification factors.

$$CVI = \sum Xi \times Ni \qquad (2)$$

$X$ − individual parameter $N$ − weight assigned to the parameter

In the present research, it was applied the weighted linear combination and the square root of the geometric mean for the final calculation. From these two approaches, the one that best represented the reality of the study area was the square root of the geometric mean. There are a large variety of models for predicting and identifying vulnerability to coastal erosion [26,53,54]. Several parameter combination approaches were used to determine the overall vulnerability of the study area, adapting the models used by [26,47,53]. In a premilitary study (data not shown), these models showed a large discrepancy with regard to the allocation of vulnerability levels. For this reason, the approach proposed by [48] was selected for the present study, with CVI taken as the square root of the geometric mean, representing the final vulnerability index. The inclusion of other important parameters, including physical processes, as the wind-wave height range was not possible, at this step, due to data availability, comparability and quality limitations, during the study period.

**3. Results**

The spatial distribution of vulnerability of the study area was described and represented, for each parameter, in the Figure 2.

Grouping all vulnerability maps by applying the geometric mean square root, resulted in the global vulnerability map (CVI) of the study area (Figure 3). The final result was obtained after reclassifying all parameters of parameters based on intervals of quartiles

and visualization data, divided into five levels. Approximately 1220 km² of the study area were evaluated towards the vulnerability to coastal erosion; the integrated table in the Figure 3. demonstrated the vulnerability associated with each area.

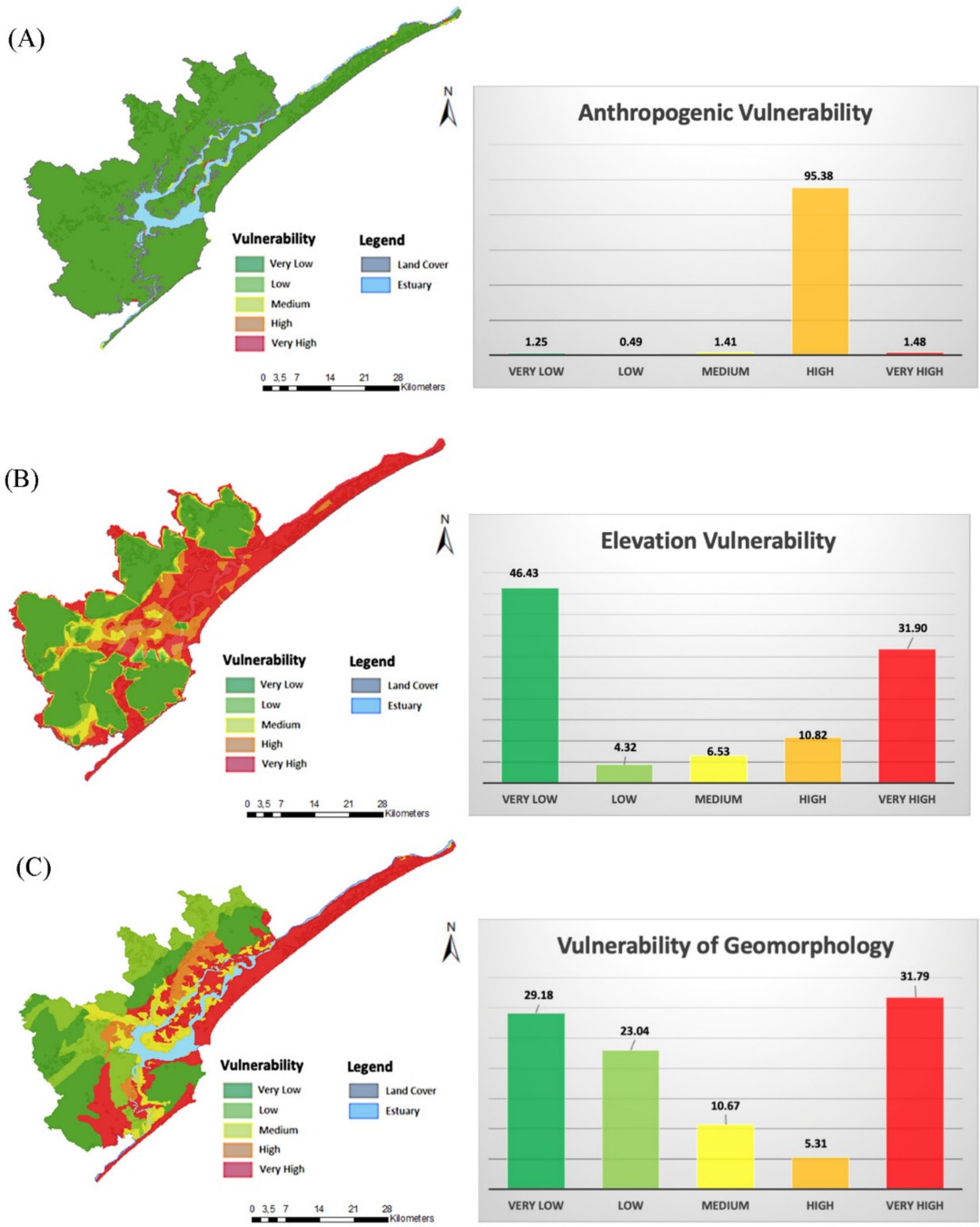

**Figure 2.** *Cont.*

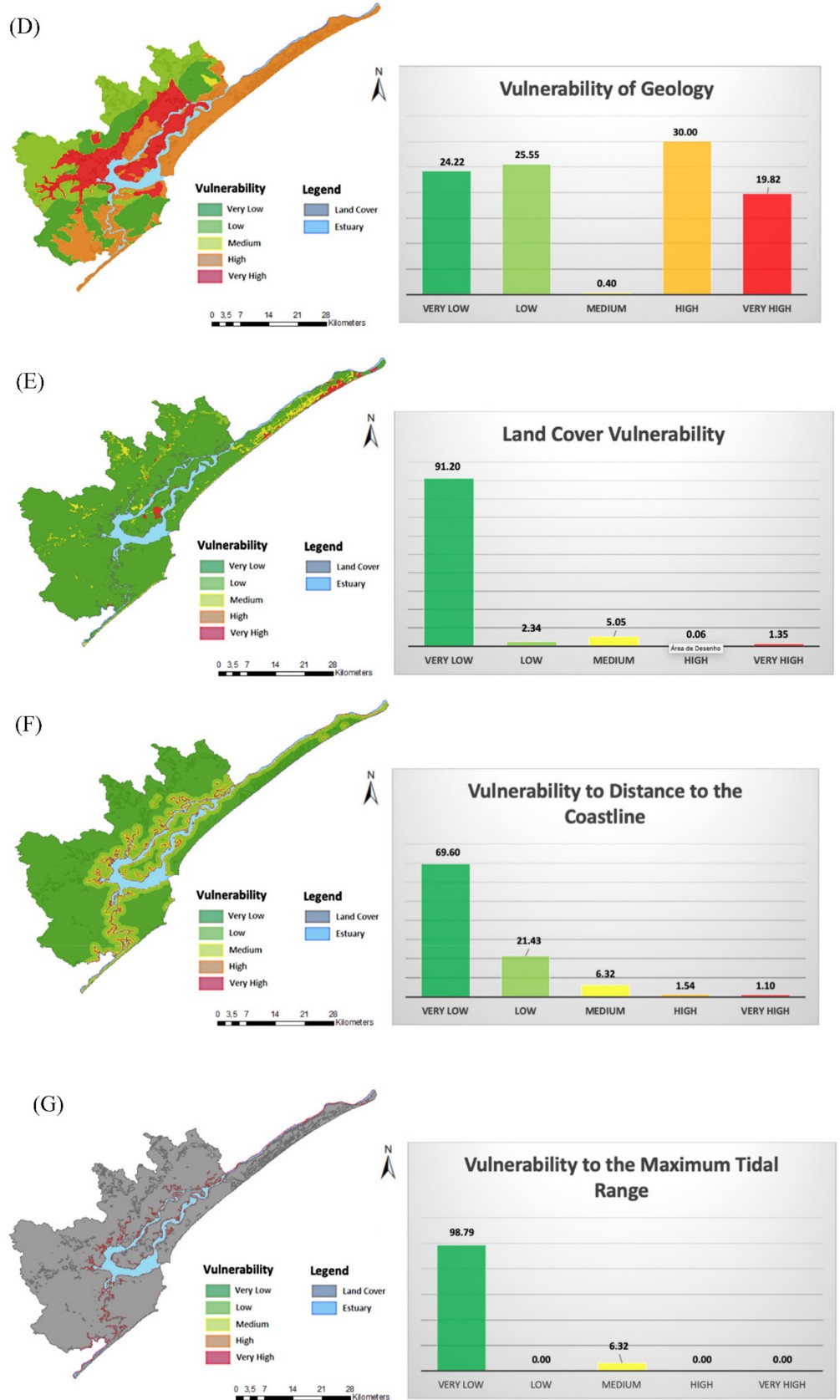

**Figure 2.** Modelled vulnerability for (**A**) anthropogenic activities, (**B**) elevation, (**C**) geomorphology, (**D**) geology, (**E**) land cover, (**F**) distance to the coastline and (**G**) maximum tidal range. The vulnerability levels associated with each area as well as its corresponding percentage (in the integrated graphic) are also represented.

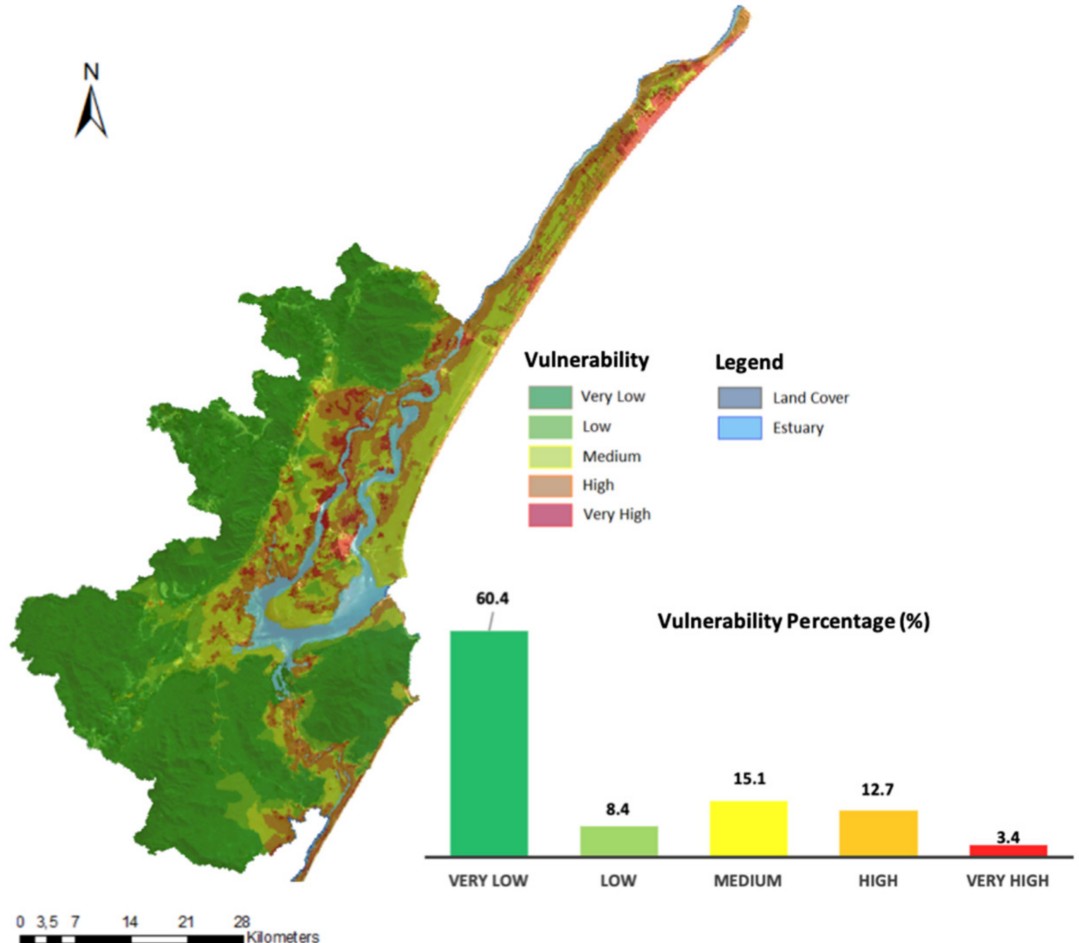

**Figure 3.** Erosion Global vulnerability levels of the Cananéia-Iguape system (24°50′, 25°40′ S; 47°20′, 48°20′ W). Vulnerability Areas (Vulnerability percentage associated with each area as well as its corresponding percentage) are also represented.

The CVI values ranged between 0 and ≈82. CVI values below 1.66 were assigned to the very low vulnerability category. Values from 1.67 to 3.32 were assigned to the low vulnerability category. Values from 3.33 to 4.98 were assigned to medium vulnerability. Values that ranged from 4.99 to 6.64 were considered as high vulnerability. The last category (very high) comprises values between 6.65 to 82. In this category, the value 82 was added to the penultimate value (8.3) due to pixel interference in the final result, because this value leaves out the rating scale of 6.65 to 8.3. Approximately 1221 km$^2$ were evaluated. Of this total 12.7% (155 Km$^2$) were classified as highly vulnerable and 3.4% (41 Km$^2$) as very high vulnerability. The areas vulnerability (Vulnerability representation associated with each area as well as its corresponding percentage in terms of Area (Km$^2$) and Percentage (%)) were: Very low-737 Km$^2$, representing 60.4%; Low-102.9 Km$^2$; representing 8.4%; Medium-184.7 Km$^2$, representing 15,1%; High-154.8 Km$^2$, representing 12.7%; Very high-40.9 Km$^2$, representing 3.4% (Figure 3). It should be noted that the data collected during the fieldwork had a great influence on the result (data on vulnerability to erosion areas), resulting all of them in polygons classified as very high vulnerability (Figure 3).

## 4. Discussion

Both natural processes and Human activity have extensively modified mangroves, and the recent increasing economic and social activities are increasingly impacting severely these structures [4,7]. These impacts have created dramatic changes over a short period of time, prompting a large-scale disappearance of these areas [7,57]. Erosion is a worldwide process that is intensified by anthropogenic activity, resulting in a variety of impacts, with emphasis on the loss of mangrove forests and shoreline [2,4,5]. The decline of mangrove

forests and the associated erosion phenomena are now considered among the major impacts caused in coastal areas [11,12]. This phenomenon has led to significant changes in the mangrove forest with consequences on the set of activities and structures of populations across the coastal ecosystem [4,58]. Therefore, it is necessary to highlight the consequences resulting from the coastline retreat, the disappearance of beach areas, loss and imbalance of natural habitats, increase in the frequency and magnitude of floods, loss of property and public and private goods, loss of landscape value, losses in socio-economic activities and large financial losses to the recovery of coastal areas [4,8]. For this reason, there is an urgent need to identify the most affected areas and intercede in order to predict this process and preserve these unique ecosystems. Although this is not easily predictable, because of the number of factors that contributes to this process [53]. In fact, it involves several tasks and integrated processes, including remote and/or in situ monitoring, perform structured analysis of the several involved factors, develop visual representations in space and time and evaluate the changes [4,59]. The use of these representations is a method that leads to a better understanding of the spatial distribution of erosion [26], which is of extreme importance for decision-making in assessing and forecasting risks, so as to avoid damage [4].

To identify existing modifications in the coastal system of Cananéia-Iguape and to determine the overall vulnerability to erosion, it was necessary to identify a wide range of parameters and determine the best way to combine them. The parameters defined in the present work resulted in the availability of existing data but also based on defined and validated models by several authors for the study of higher impact factors for coastal erosion [46–49,51,54]. Elevation corresponded to the vulnerability of low-lying areas to the impact of waves and storms (i.e., smaller the elevation the more likely it to be eroded) [60]. In such circumstances, many areas, especially those that are clear of mangroves, are more likely to suffer from erosion [61]. While there is intensive exploitation in the mangroves of Cananeia, there were a large number of areas with relatively small elevation where these processes were observed. The obtained data in both qualitative and quantitative format, and in different scales of units, was based and obtained in literature, to assign a scale of classification to the parameters of 1 to 5, representing 1 the very low vulnerability class and 5 to very high. This approach allowed not only to classify the several parameters in the same class of values, but also simplified the representation and identification of the areas of greatest vulnerability. The geomorphology expressed erodibility on different types of terrain [62]; given the existence in the study area of several areas classified as dunes, plains and alluvial deposits in the region, the inclusion of this variable was necessary. The inclusion of geology was considered in the study, since approximately 50% of the study area was composed of unconsolidated sediments and sedimentary rocks; allowing the evaluation of sediments, directly connected to the erosive processes [48,49,51]. There were, also, identified areas where was a strong evidence of erosion by removing large areas of mangroves having consequences in the mangrove dynamics associated with human activities, and for this reason, it was introduced the parameter of anthropogenic activities. The different type of land cover has a diverse level of vulnerability to erosion, as in the of coastal areas covered by flooring, urbanizations, as the mangrove presented different morphological behaviour [51]. According to [51], the higher the level of change of the natural state of ground covering, the greater their vulnerability will be. The maximum tidal range is linked both to flooding as to the risk of erosion [48]. Although a large tidal range dissipates wave energy, limiting the beach areas to a brief low tide period it also defines a wide area of wetlands to be more susceptible to flooding [48]. The velocity of the water is greater at low tide, allowing a greater movement of sediments in flood zones [63]. When the exposure index through wave impact is combined with low-lying areas, it provides a coastal vulnerability indicator [55]. The selection of the distance to the coastline, also, proved to be determinant for the analyses. According to [21], vulnerability to erosion in coastal regions increases with proximity to the interaction zone with the sea, getting these most exposed to its action. The application of the square root of the

geometric mean, although of greater simplicity, showed a greater similarity with the reality of the erosive processes in the study area, presenting a good tool for the vulnerability to erosion evaluation in the different worldwide mangroves. The implementation of the adapted model for the study area showed a homogeneous distribution of vulnerability in the Cananéia-Iguape estuarine system.

Stand out the greater the number of interventions by man in a given area, the greater the deterioration of coastal erosion [51,64]. The coastline of Cananeia island and the north of Ilha Comprida were examples of constant anthropogenic interference areas, which had higher vulnerability levels, mainly because these areas are the centre of activity in the region and the results. Predicted another important aspect to emphasize was the number of interventions in mangrove areas that took place in the mangroves of Cananeia, many of these areas that were analysed during fieldwork were found occupied by infrastructures, with evident and important signs of erosion. Through observation of the final result of the CVI, it was verified that there was a great area spot classified as very low vulnerability. In fact, about 91% of the selected area is occupied by forest, both halophyte and ombrophilous, and to a certain extent a positive aspect with regard to erosion prevention, since this type of forest plays an important role in erosion prevention in coastal regions [37]. The results also highlighted the fact that the areas classified as high and very high vulnerability were significantly influenced by anthropogenic activities and land occupation, especially urbanization and the set of anthropogenic interventions in areas of small non-consolidated sediments, dune and plains areas, being these more vulnerable to the erosive processes. The coastline of the island of Cananéia and the north of the Ilha Comprida were relatively sensitive areas, which had higher vulnerability ratings. This was mainly because that these areas are the centre of the anthropogenic activity of the region, thus predicting such results.

## 5. Conclusions

Geographic Information Systems, remote sensing and mapping can represent an important contribute to geographic and spatial aspects of mangroves vulnerability and coastal erosion assessment in order to facilitate decision-making and management of coastal areas. The results obtained, namely the coastal vulnerability index, provided a sample of the reality observed in the study area, which, together with the database. The selected approach showed to be a good tool for the vulnerability to erosion evaluation. The implementation of the adapted model for the study area showed a homogeneous distribution of vulnerability in the Cananéia-Iguape estuarine system. In addition, particular attention should be given to the increase of anthropogenic interferences that threatens the survival of mangroves in this region, with a significant contribution to erosive processes. The results revealed the urgent need to monitor regularly the areas where these processes take place in order to analyse several related factors, develop various types of visual representations of different times and temporal scales to assess and predict changes that may occur in the future. Predicting the behaviour of erosion can avoid the increase of vulnerability and erosion rates and reduce the damage to the ecosystem and the economy of the region. Through GIS it was possible to accomplish mangroves vulnerability and coastal erosion assessment on different extension levels, with an accurate analysis of the environmental phenomena, although it could be optimized when GIS is combined with Remote Sensing data. The advantages of this combination have already been described to the creation of more embracing maps of the distribution of species, analysis of ecosystems and landscapes, change of climatic conditions and invasiveness issues.

**Author Contributions:** Conceptualization: Fernando Morgado, José Guilherme Vieira and Edison Barbieri; Methodology: Fernando Morgado, José Guilherme Vieira and Edison Barbieri; Software: José Guilherme Vieira; Validation: Fernando Morgado, José Guilherme Vieira and Edison Barbieri; Formal analysis: Luís Russo Vieira, José Guilherme Vieira, Isabel Marques da Silva, Edison Barbieri and Fernando Morgado; Investigation: Luís Russo Vieira, José Guilherme Vieira, Isabel Marques da Silva, Edison Barbieri and Fernando Morgado; Writing—original draft preparation: Luís Russo Vieira, José Guilherme Vieira, Isabel Marques da Silva, Edison Barbieri and Fernando Morgado;

Writing—review and editing: Luís Russo Vieira, José Guilherme Vieira; Isabel Marques da Silva; Edison Barbieri and Fernando Morgado: Visualization, Luís Russo Vieira, José Guilherme Vieira, Isabel Marques da Silva; Edison Barbieri and Fernando Morgado. All authors have read and agreed to the published version of the manuscript.

**Funding:** This work was supported by the Biology Department and by the Centre for Environmental and Marine Studies (CESAM) from University of Aveiro funds, co-financed by QREN, Programa Operacional Regional do Centro e União Europeia/Fundo Europeu de Desenvolvimento Regional. The authors would like to thank the Center for Research and Environmental Conservation from the Faculty of Natural Sciences (CICA), Faculty of Natural Sciences (FCN) of the University Lúrio (UniLúrio), Mozambique, for financial support of this paper.

**Institutional Review Board Statement:** Not applicable.

**Informed Consent Statement:** Not applicable.

**Data Availability Statement:** Not applicable.

**Conflicts of Interest:** The authors declare no conflict of interest.

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
