# Peer review of "GIS Models for Vulnerability of Coastal Erosion Assessment in a Tropical Protected Area"

_ijgi, doi:10.3390/ijgi10090598_

Round 1
Reviewer 1 Report
This manuscript presents a nontrivial research. I recommend it for publication on the journal.
Some revision is needed. The title includes mangroves vulnerability; however, its focuses are not on mangroves. First, no single parameter is directly associated with mangroves, for example, status of mangroves. Second, only four paragraphs in the manuscripts are directly related to mangroves; they are first three paragraphs of Introduction and Section 2.2. All others are about coastal erosion assessment. “GIS models for vulnerability of coastal erosion assessment in a tropical protected area” would fit its content much better.
Author Response
Comments and Suggestions for Authors
This manuscript presents a nontrivial research. I recommend it for publication on the journal.
Some revision is needed. The title includes mangroves vulnerability; however, its focuses are not on mangroves. First, no single parameter is directly associated with mangroves, for example, status of mangroves. Second, only four paragraphs in the manuscripts are directly related to mangroves; they are first three paragraphs of Introduction and Section 2.2. All others are about coastal erosion assessment. “GIS models for vulnerability of coastal erosion assessment in a tropical protected area” would fit its content much better.
Answer to Reviewer #1 Comments: The manuscript title was changed to “GIS models for vulnerability of coastal erosion assessment in a tropical protected area” as suggested.

Reviewer 2 Report
This study focuses on an important topic of conservation of the natural environment, for which GIS technologies are fundamental for the production of useful cartographic documents for the study of the evolution of degradation in mangrove zones and protected forest areas and how these constitute important management and decision support tools.
Overall the document is well structured and well written. It is easy to read and understandable in the methodology applied.
Three aspects that I believe can be changed/improved:
- The title of the work does not exactly correspond to the study carried out; the authors do not explore so many vulnerability models or even compare their results, much less validate them; as such the title should not be focused on "GIS models" but more on "vulnerability assessment ... based on GIS" or " Vulnerability index ... to estimate…." something like this.
- Although the authors mention that the data was acquired from the Water Resources Fund of São Paulo, it seems to me essential to include a section to describe the metadata of each layer of information used (scale/resolution, source, reference system, date of production, etc.); It is not understood where remote sensing is use dor applied - can the authors better explain which data is used and its characteristics?
Data can strongly condition the results from the point of view of the positional rigor, mainly because the authors intend to analyze % of the occupation area of the various classes of vulnerability!
- I am not an expert in the field of conservation or in what concerns the study of mangroves or forests, but I realize that the model used is too simple, perhaps unrepresentative of reality. The fact that the parameters have the same weight on the model seems unrealistic to me. The altitude has the same importance for this study as, for example, the distance to the coastline or the land cover? maybe not!!! - I think the authors do not defend their model option very well!
Minor aspects of form:
Line 152: full stop to remove (?)
Line 176: must include the longitude range as well
Line 214: replace , by . after [45]
Figure 1.: Legend must describe what isolines mean (altitude?)
Line 258: “The application of the GIS models…” – no need again to describe in full GIS
Lines 310, 316: maybe the math formulas have to be numbered
Figure 3: At least the map of Figure 3 should have a grid with coordinates in an indicated reference system, so that it is possible to know (in the exact coordinates or near) where the very high vulnerability sites or other are located; If GIS is used and the data are georeferenced, it makes no sense to present a map without knowing "where" a given result is located. That's what the coordinates are for!
Author Response
This study focuses on an important topic of conservation of the natural environment, for which GIS technologies are fundamental for the production of useful cartographic documents for the study of the evolution of degradation in mangrove zones and protected forest areas and how these constitute important management and decision support tools.
Overall the document is well structured and well written. It is easy to read and understandable in the methodology applied.
Three aspects that I believe can be changed/improved:
The title of the work does not exactly correspond to the study carried out; the authors do not explore so many vulnerability models or even compare their results, much less validate them; as such the title should not be focused on "GIS models" but more on "vulnerability assessment ... based on GIS" or " Vulnerability index ... to estimate…." something like this.
- The manuscript title was changed to “GIS models for vulnerability of coastal erosion assessment in a tropical protected area”. The Abstract and Introduction sections were, also, revised focusing on vulnerability assessment.
Although the authors mention that the data was acquired from the Water Resources Fund of São Paulo, it seems to me essential to include a section to describe the metadata of each layer of information used (scale/resolution, source, reference system, date of production, etc.); It is not understood where remote sensing is use dor applied - can the authors better explain which data is used and its characteristics? Data can strongly condition the results from the point of view of the positional rigor, mainly because the authors intend to analyze % of the occupation area of the various classes of vulnerability! I am not an expert in the field of conservation or in what concerns the study of mangroves or forests, but I realize that the model used is too simple, perhaps unrepresentative of reality. The fact that the parameters have the same weight on the model seems unrealistic to me. The altitude has the same importance for this study as, for example, the distance to the coastline or the land cover? maybe not!!! - I think the authors do not defend their model option very well!
- This work represents a first application of an adapted model, based on CVI, in the estuarine system of Cananeia-Iguape (Brazil), a tropical protected area. The coastal vulnerability index is a popular index in literature to assess the coastal vulnerability, widely used in several studies, being based on key-parameters for a first vulnerability assessment, including elevation, geomorphology, geology, land cover, anthropogenic activities, distance to the coastline, and maximum tidal range. The parameters do not all have the same weight, as indicated in formula 2 (N-weight assigned to the parameter). This section has also been revised and clarified. Additionally, new information has been included to support of this model option in the section 2.5.
Minor aspects of form:
Line 152: full stop to remove (?)
- Full stop was removed
Line 176: must include the longitude range as well
- The longitude range was included.
Line 214: replace , by . after [45]
- The full stop was included after [45]
Figure 1.: Legend must describe what isolines mean (altitude?).
- The figure 1 was replaced, as suggested by the reviewers.
Line 258: “The application of the GIS models…” – no need again to describe in full GIS
- The full description was deleted.
Lines 310, 316: maybe the math formulas have to be numbered
- Both formulas were numbered.
Figure 3: At least the map of Figure 3 should have a grid with coordinates in an indicated reference system, so that it is possible to know (in the exact coordinates or near) where the very high vulnerability sites or other are located; If GIS is used and the data are georeferenced, it makes no sense to present a map without knowing "where" a given result is located. That's what the coordinates are for!
- Considering the composition of the figure, with the graphic included, the coordinates were described in the legend.

Reviewer 3 Report
The paper describes the coastal erosion assessment of the Cananéia-Iguape estuarine system, with particular reference to the vulnerability of the mangroves ecosystem.
I am not sure about the originality and novelty of this work. The manuscript presents (only?) an application of a (well-known) methodology (CVI index), without given nor some consideration of its applicability (e.g. in terms of parameter uncertainties), nor some implications on the findings in the face of potential climate change and sea-level rise.
Moreover, the authors did not include in the analysis the effect of wind waves, which play a key role in every coastal erosion study. I encourage the authors to delve into the literature on the subject which is a primary requirement for every study.
Therefore, I suggest revising broadly the manuscript, in order to include all the essential parameters. In the following, I list some suggestions to revise the manuscript.
1) The abstract should be revised since it should be a concise and comprehensive reflection of what is in the paper. The first three sentences of the abstract are too general and focused on the broad topic of coastal erosion. However, the study is mainly referred to the vulnerability of Mangroves ecosystem. I suggest leaving only a single general sentence and then a focus on your methodology and the peculiarities of the study.
2) The keywords should not be repeated in the title and thus, I suggest replacing “Coastal erosion”, “Mangroves” and “Vulnerability”.
3) I feel the introduction is too long. I suggest reviewing it, trying to keep only the aspects that are relevant for the study. For instance, both the paragraphs from lines 69 – 99 (importance of mangrove forests) and from lines 103 – 120 (GIS advantages) should be more concise. Moreover, the objectives of the research are restricted in the last paragraph, without a description of the peculiarities of the present study.
4) Line 91: Together with reference [19], add the recent SROCC (2019) “Special report on the ocean and cryosphere in a changing climate” Intergovernmental Panel on Climate Change (IPCC). (see in particular Chapters 4, 5 and 6).
5) Line 99: “Astronomical damage with the recovery of coastal areas”. Please rephrase since I do not understand what is “astronomical damage”.
6) Figure 1 is ineffective. It is very difficult for an international reader to follow your description and to understand what the main features of the studied area are. All the polygons are green, but I think that some polygons are barrier islands and other lagoons and channels.
7) As I mentioned at the beginning, the authors did not include the wind-wave as a parameter for the assessment of coastal erosion.
The authors explain only the tidal range (line 198), however, the investigated area includes both the inner part of the estuarine-lagoon system and the barrier islands that face the Atlantic Ocean. Therefore, it is essential to consider the wave climate. In fact, the wave climate is the major responsible for coastal erosion in low-lying coastal areas. I suggest adding a broad description of the main marine forcing conditions (winds, tides, storm surges, wind waves, etc.).
This lack is also included in the choice of the vulnerability parameters. Usually, the CVI index includes also the typical wind-wave height range (see Koroglu et al. 2019, Pantusa et al. 2018). The wave characteristics are essential for a coastal vulnerability study. In fact, I expect that all the barrier islands facing the ocean and exposed to wind waves will have higher values of CVI.
Moreover, in line 486, the authors explain that the area with lower elevations are exposed to the impact of waves and storms, without referring to this issue along the paper.
Koroglu, A., Ranasinghe, R., Jiménez, J. A., & Dastgheib, A. (2019). Comparison of coastal vulnerability index applications for Barcelona Province. Ocean & coastal management, 178, 104799.
Pantusa, D., D’Alessandro, F., Riefolo, L., Principato, F., & Tomasicchio, G. R. (2018). Application of a coastal vulnerability index. A case study along the Apulian Coastline, Italy. Water, 10(9), 1218.
Author Response
The paper describes the coastal erosion assessment of the Cananéia-Iguape estuarine system, with particular reference to the vulnerability of the mangroves ecosystem.
I am not sure about the originality and novelty of this work. The manuscript presents (only?) an application of a (well-known) methodology (CVI index), without given nor some consideration of its applicability (e.g. in terms of parameter uncertainties), nor some implications on the findings in the face of potential climate change and sea-level rise.
- This work represents a first application of an adapted model, based on CVI, in the estuarine system of Cananéia-Iguape (Brazil), a tropical protected area. The coastal vulnerability index is a popular index in literature to assess the coastal vulnerability, widely used in several studies, being based on key-parameters for a first vulnerability assessment, including elevation, geomorphology, geology, land cover, anthropogenic activities, distance to the coastline, and maximum tidal range. This section has also been revised and clarified. New information has been included to support of this model option in the section 2.5.
Moreover, the authors did not include in the analysis the effect of wind waves, which play a key role in every coastal erosion study. I encourage the authors to delve into the literature on the subject which is a primary requirement for every study. Therefore, I suggest revising broadly the manuscript, in order to include all the essential parameters. In the following, I list some suggestions to revise the manuscript.
1) The abstract should be revised since it should be a concise and comprehensive reflection of what is in the paper. The first three sentences of the abstract are too general and focused on the broad topic of coastal erosion. However, the study is mainly referred to the vulnerability of Mangroves ecosystem. I suggest leaving only a single general sentence and then a focus on your methodology and the peculiarities of the study.
- The abstract section was rewritten, with a major focus on coastal erosion.
2) The keywords should not be repeated in the title and thus, I suggest replacing “Coastal erosion”, “Mangroves” and “Vulnerability”.
- The Keywords were changed, reflecting the study main focus.
3) I feel the introduction is too long. I suggest reviewing it, trying to keep only the aspects that are relevant for the study. For instance, both the paragraphs from lines 69 – 99 (importance of mangrove forests) and from lines 103 – 120 (GIS advantages) should be more concise. Moreover, the objectives of the research are restricted in the last paragraph, without a description of the peculiarities of the present study.
- The Introduction was revised and simplified.
4) Line 91: Together with reference [19], add the recent SROCC (2019) “Special report on the ocean and cryosphere in a changing climate” Intergovernmental Panel on Climate Change (IPCC). (see in particular Chapters 4, 5 and 6).
- The suggested reference was included in the text. Contributions from Chapters 4,5 and 6 from SROCC (2019) were also included in the text.
5) Line 99: “Astronomical damage with the recovery of coastal areas”. Please rephrase since I do not understand what is “astronomical damage”.
- The sentence was revised.
6) Figure 1 is ineffective. It is very difficult for an international reader to follow your description and to understand what the main features of the studied area are. All the polygons are green, but I think that some polygons are barrier islands and other lagoons and channels.
- The Figure 1 was replaced.
7) As I mentioned at the beginning, the authors did not include the wind-wave as a parameter for the assessment of coastal erosion.
The authors explain only the tidal range (line 198), however, the investigated area includes both the inner part of the estuarine-lagoon system and the barrier islands that face the Atlantic Ocean. Therefore, it is essential to consider the wave climate. In fact, the wave climate is the major responsible for coastal erosion in low-lying coastal areas. I suggest adding a broad description of the main marine forcing conditions (winds, tides, storm surges, wind waves, etc.).
This lack is also included in the choice of the vulnerability parameters. Usually, the CVI index includes also the typical wind-wave height range (see Koroglu et al. 2019, Pantusa et al. 2018). The wave characteristics are essential for a coastal vulnerability study. In fact, I expect that all the barrier islands facing the ocean and exposed to wind waves will have higher values of CVI.
Moreover, in line 486, the authors explain that the area with lower elevations are exposed to the impact of waves and storms, without referring to this issue along the paper.
- The CVI index is a widely used to assess the coastal vulnerability. It is a very flexible index, based on key parameters, that we believe to be included in the present work. In addition, in a premilitary study all parameters were tested, before being included in the model. The inclusion of other important parameters- physical processes, as the wind-wave height range was not possible, at this step, due to data availability, comparability and quality limitations, during the study period. The application of the chosen model, although of greater simplicity, showed a greater similarity with the reality of the erosive processes in the study area, presenting a good tool for the vulnerability to erosion evaluation.

Round 2
Reviewer 3 Report
Thank you for your edits and replies in response to my comments.
I now find the manuscript suitable for publication.
I only suggest adding a sentence that explains why you don't consider wind waves in your vulnerability analysis (as you write in the answer to the reviewer: "The inclusion of other important parameters- physical processes, as the wind-wave height range was not possible, at this step, due to data availability, comparability, and quality limitations, during the study period.")
Author Response
We would like to thank both Editor and Reviewers for allowing us to revise and resubmit the present manuscript. The manuscript was carefully revised, including the Reviewer #3 comments and suggestions, as requested. In addition, other minor corrections were made to improve the text. All revisions made were marked (Track Changes).
We are very grateful to the reviewers for their time, their careful reading and, their helpful comments and recommendations. We hope that the revised version of the manuscript can be now considered as suitable for publication.
Answer to Reviewer #3 Comments: A sentence, explaining why the wind waves were not possible to be considered in the vulnerability analysis, was included at the end of section 2.5. (Lines 336-339).
